# Comparative Analysis of Whitening Outcomes of Over-the-Counter Toothpastes: An In Vitro Study

**DOI:** 10.3390/dj13020045

**Published:** 2025-01-22

**Authors:** Md Sofiqul Islam, Vivek Padmanabhan, Kamar Ali Shanati, Ahmed Malalah Naser, Nada Tawfig Hashim, Smriti Aryal A C

**Affiliations:** 1Department of Operative Dentistry, RAK College of Dental Sciences, RAK Medical and Health Sciences University, Ras Al-Khaimah P.O. Box 12973, United Arab Emirates; 2Department of Pediatric Dentistry, RAK College of Dental Sciences, RAK Medical and Health Sciences University, Ras Al-Khaimah P.O. Box 12973, United Arab Emirates; vivek.padmanabhan@rakmhsu.ac.ae; 3Department of Periodontology, RAK College of Dental Sciences, RAK Medical and Health Sciences University, Ras Al-Khaimah P.O. Box 12973, United Arab Emirates; nada.tawfig@rakmhsu.ac.ae; 4Department of Oral and Craniofacial Health Sciences, College of Dental Medicine, University of Sharjah, Sharjah P.O. Box 27272, United Arab Emirates; saryalac@sharjah.ac.ae

**Keywords:** activated charcoal, blue covarine, carbamide peroxide, CIELab, hydrogen peroxide, sodium tripolyphosphate, staining–brushing cycles, teeth whitening, titanium dioxide

## Abstract

**Background/Objectives**: Whiter teeth are widely accepted as the most beautiful. With the growing demand for whiter teeth, several manufacturers have launched different brands of whitening toothpaste claiming to be effective in removing tooth stains and whitening teeth. The objective of this in vitro study was to evaluate and compare the whitening effect of eight over-the-counter available toothpastes by measuring the changes in color using a digital colorimeter in a simulated staining–brushing cycle model. **Methods**: A total of 32 extracted bovine enamel were polished with 1000–2000 grit SIC paper and immersed in a tea staining solution for 30 min at 37 °C to create extrinsic staining. The specimens were randomly divided into eight groups (*n* = 4) and subjected to a staining–brushing cycle for 2 weeks. During this period, the specimens were stained for 5 min twice, brushed for 2 min twice per day, and immersed in artificial saliva for the remaining time. The colors of the enamel surfaces L*, a*, and b* were recorded, and the color difference (ΔE) was measured before the treatment and after 1 and 2 weeks of the staining–brushing cycle. **Results**: The repeated measures showed a significant reduction of (ΔE) after 1 week of the staining–brushing cycle in all eight experimental groups (*p* < 0.05). The (ΔE) was significantly reduced after the second week of the staining–brushing cycle in groups 1, 3, and 5 (*p* < 0.05). However, it was insignificant in groups 2, 6, 7, and 8 (*p* > 0.05). Among the tested materials, group 1 showed the highest and group 8 showed the lowest teeth-whitening effects. **Conclusions:** The over-the-counter toothpaste used in this study showed effective teeth-whitening. Charcoal-based toothpaste showed the highest efficiency in teeth whitening.

## 1. Introduction

The color of a person’s teeth has an instant impact on their perceived beauty, personality, and overall quality of life [1]. The concept of white and bright teeth is often associated with youth, health, and attractiveness in modern society [2]. Social experiment results have shown that people with whiter teeth are generally perceived as more confident, successful, and socially appealing [3,4]. A poor appearance of teeth has a ripple effect on mental well-being, making individuals more self-conscious and reducing their overall quality of life [5]. The tooth color of an individual is intricately linked to the esthetic component of their facial beauty. Previous research reports have indicated that esthetic dental procedures, like teeth whitening, can increase the social interactions, sense of confidence, and personal relationships of individuals [6,7].

Enamel, the outmost layer of the tooth, plays a key role in the overall appearance of teeth. The main contribution of enamel to color is the degree of opacity or translucency. This characteristic of enamel, along with its thickness, determines the amount of dentin that affects the overall appearance of the tooth [8,9]. Unlike enamel, dentin has a more sustainable and opaque structure, giving it a yellowish to light brown color. The color of dentin is influenced by hydroxyapatite concertation, collagen fiber, and water, which interacts with light to produce a saturated hue [10].

Regardless of the color of a tooth after development, it can undergo discoloration due to extrinsic and intrinsic factors [11]. The outer surface of the enamel’s color can be altered by diet, oral hygiene, and personal habits. Food and drink, like coffee, tea, and spices, can stain enamel over time. Smoking and chewing tobacco are also major contributors to extrinsic stains. Additionally, intrinsic stains can originate within the tooth structure due to trauma, certain medications, systemic conditions, and aging. Dental trauma that affects the blood supply to a tooth can also result in intrinsic discoloration, usually presenting as a dark or black tooth. Aging naturally leads to changes in tooth color as the enamel wears down, making the yellowish dentin more visible [12].

Teeth whitening is a cosmetic dental procedure performed to improve the appearance of teeth by removing stains and discoloration. Teeth whitening is a non-invasive, cost-effective treatment that can be done professionally in dental offices or through at-home products. In-office whitening or bleaching is a professionally preformed clinical procedure that typically involves the application of higher concentrations of bleaching agents directly on the tooth surface, allowing for faster and more significant results [13]. At-home teeth whitening or home bleaching procedures are usually performed using a customized tray or strip using a lower concentration of bleaching gel prescribed by the dentist. Hydrogen peroxide and carbamide peroxide are the main ingredients used in both in-office and at-home bleaching gel [14]. Based on the patient’s preference and expected outcomes, both procedures have been proven to be effective in improving tooth color.

To cope with the growing demand for teeth whitening, over-the-counter tooth-whitening products in the form of toothpaste have been brought to the market by several manufacturers [15]. The promotional advertisements of these toothpastes on TV channels and internet platforms have made them attractive to customers. The currently available over-the-counter whitening toothpastes utilize various active ingredients to remove surface stains and enhance tooth brightness. The key components included in these toothpastes are either an abrasive substance, like hydrated silica, calcium carbonate, or magnesium carbonate, or chemicals, like hydrogen peroxide, carbamide peroxide, activated charcoal, blue covarine, or a combination of these. These ingredients function through mechanical abrasion, chemical bleaching, or optical modification to achieve a whitening effect [16,17]. However, the use of such ingredients can negatively affect the enamel and periodontium. Moreover, the effectiveness of these toothpastes in clinical and laboratory setups has not been well documented [18,19]. Therefore, it is essential to evaluate the effectiveness of these toothpastes on teeth whitening for clinical recommendations and public awareness.

A laboratory or simulation-based pre-clinical study is an effective way to predict the clinical outcome of a certain product or procedure. This kind of setup can eliminate several factors that might directly or indirectly influence the research outcomes of a clinical study due to variations among human subjects. Additionally, it can prevent and/or protect the human subject from the possible side effects of materials [20].

The aim of this in vitro study was to evaluate the effectiveness of tooth-whitening toothpastes currently available in the market. The objective was to evaluate and compare the whitening effects of eight over-the-counter available toothpastes by measuring the changes in color using a digital colorimeter in a simulated staining–brushing cycle model.

## 2. Materials and Methods

**Study design:** This study’s in vitro protocol was reviewed and approved by the RAK Medical and Health Sciences University research and ethical committee, with approval number RAKMHSU-REC-011-2022/23-UG-D.

**Sample size calculation:** The number of required samples for this study was calculated using statistical software G*Power 3.1.9.7. With an effect size of 0.80, a confidence level of 95%, and an estimated power of 0.95, the required number of specimens for each group was calculated to be *n* = 4 for this repeated measures experiment.

**Specimen preparation:** A total of 32 extracted bovine incisor teeth free from cracks and discoloration were used in this pre-clinical study. The teeth were kept frozen at −50 °C from the time of collection until the experiment. Then, they were thawed in running tap water, and the soft tissue remnants were gently cleaned off using a scalpel. The enamel was polished using a series of 1000–2000-grit silicon carbide (SiC) papers to obtain a flat and smooth surface. The surfaces were observed using magnification to ensure the uniformity of the surfaces and the elimination of scratches. After rinsing them with tap water and letting them air dry, the color of the enamel surfaces was measured with a dental colorimeter (VITA Easy Shade, VITA Zahnfabrik, Bad Säckingen, Germany), and the CIE L*, C*, H*, a*, and b* values were recorded. In order to reduce artifacts, the reading of each specimen was taken 3 times following the guidelines given by Islam MS et al., 2024 [21]. The data were recorded as the baseline tooth color (B), with the color parameters labeled as L*B, C*B, H*B, a*B, and b*B.

**Staining of the specimens:** The staining and color parameter measurements were conducted following the procedure in a previously published article [22]. The staining solution was prepared by immersing two tea bags in 50 mL of boiling water for 5 min. The specimens were immersed in the staining solution and stored in an incubator for 30 min at 37 °C. After rinsing them with tap water and letting them air dry, the L*, C*, H*, a*, and b* color values of the enamel surfaces were recorded 3 times with a dental colorimeter. The averages of 3 measurements were recorded as the stained tooth color (S), with the color parameters labeled as L*S, C*S, H*S, a*S, and b*S. The difference in color (tooth discoloration) after staining (ΔS) was calculated using the CIELab formula [23]:$$\Delta S_{ab}={\left(\Delta L^{*2}+\Delta a^{*2}+\Delta b^{*2}\right)}^{1/2}$$
where ΔL = L*S − L*B, Δa = a*S − a*B, and Δb = b*S − b*B.

**Simulated staining and brushing cycle:** After staining, the specimens were randomly divided into 8 groups (*n* = 4) using a manual randomizer. The specimens were then subjected to a simulated staining–brushing cycle for 2 weeks. During this cycle, the specimens were stained for 5 min, followed by brushing with one of the whitening toothpastes listed in Table 1 using a power toothbrush for 2 min at a constant speed (approximately 31,000 brush strokes per minute) and pressure (250 gm) to complete 1 cycle. The staining–brushing cycle was performed twice a day, and during the remaining time, the specimens were immersed in artificial saliva. The artificial saliva contained 0.7 mM CaCl_2_, 0.2 mM MgCl_2_·6H_2_O, 4.0 mM KH_2_PO_4_, 30 mM KCl, 0.3 mM NaN_3_, and 20 mM HEPES buffer [24]. The color parameters L*, C*, H*, a*, and b* were re-evaluated after 1 week and 2 weeks of the staining–brushing cycle. The averages of 3 measurements of each specimen were recorded as the whitened tooth color (W), with the color parameters labeled as L*W, C*W, H*W, a*W, and b*W. The whitening effects after 1 week and 2 weeks of the staining–brushing cycles were determined by measuring the color difference (ΔW), calculated using the following CIELab formula:$$\Delta W_{ab}={\left(\Delta L^{*2}+\Delta a^{*2}+\Delta b^{*2}\right)}^{1/2}$$
where ΔL = L*S − L*W, Δa = a*S − a*W, and Δb = b*S − b*W.

The overall whitening effect of each toothpaste after 2 weeks of the staining–brushing cycle was calculated by comparing the ΔW with the corresponding ΔS value.

**Data analysis:** The data were analyzed using statistical software (SPSS 24.0, IBM, Armonk, NY, USA). Bar graphs were generated using GraphPad Prism 10.2.3. A descriptive analysis was performed to evaluate the data distribution. The normality of the data was evaluated using the Shapiro–Wilk test and the Kolmogorov–Smirnov test. Parametric analysis was employed based on the data distribution results. Repeated-measures ANOVA was performed to evaluate the effects of staining and brushing on tooth color. Multiple comparisons among the tested groups were performed using one-way ANOVA and Tukey’s post hoc test at a 95% confidence level.

## 3. Results

Repeated-measures ANOVA showed a statistically significant effect of staining and brushing on the tooth color (*p* < 0.05). Immersion in the tea staining solution to create extrinsic discoloration significantly increased the ΔS compared to the baseline tooth color ΔB (*p* = 0.001). The ΔW was significantly reduced after 1 week (*p* = 0.001) and 2 weeks (*p* = 0.001) of the staining–brushing cycle compared to the ΔS, highlighting the whitening potential of the tested toothpastes.

The teeth specimens treated in groups 1, 3, 4, and 5 showed a significant reduction in the ΔW after the first week, which was significantly reduced even further after second week of the staining–brushing cycle (*p* < 0.05). The teeth specimens treated in groups 2, 6, 7, and 8 showed a significant reduction in the ΔW after the first week of the staining–brushing cycle (*p* < 0.05). However, the reduction in the ΔW after the second week of the staining–brushing cycle was statistically insignificant (*p* > 0.05). The changes in color after staining, after 1 week of the staining–brushing cycle, and after 2 weeks of the staining–brushing cycle using each toothpaste are shown in Figure 1, Figure 2, Figure 3, Figure 4, Figure 5, Figure 6, Figure 7 and Figure 8 and Table 2.

One-way ANOVA and Tukey’s post hoc test showed statistically significant differences in the whitening effects among the tested groups (*p* = 0.001). The Herbal Expert whitening (group 1) showed the highest whitening effect, followed by Colgate Optic White Expert (group 4), Colgate Optic White Expert Complete (group 5), Crest 3D White Deluxe (group 3), Closeup White Now Gold (group 2), Sensodyne True White (group 6), and Closeup White Now (group 7). The least whitening effect was observed in group 8, which was treated with Crest 3D White Pearl Glow. The whitening effects of all experimental groups are shown in Figure 9.

## 4. Discussion

In the current study, we attempted to simulate oral conditions by using the staining–brushing cycle method. Staining with a tea-based discoloring solution mimics the daily consumption of 2 cups of tea/coffee, with an average consumption period of 5 min. Brushing twice per day is recommended for routine dental care. For the remaining time, the specimens were incubated in artificial saliva at 37 degrees centigrade. Although an in vitro setting may not fully represent an intra-oral environment, it can closely mimic the conditions needed to study the effects of certain materials, helping to avoid adverse effects that might occur in in vivo conditions from the tested materials.

In our study, the activated charcoal-containing toothpaste Herbal Expert Whitening (group 1) showed the greatest teeth whitening effect. The popularity of activated charcoal as a personal care product ingredient has increased in recent years [25]. Activated charcoal is produced by heating carbon-rich materials in the absence of oxygen, like wood or coconut shells, resulting in a fine black powder with microscopic pores. The presence of these pores in activated charcoal helps to trap and bind chemicals and toxins. The ability to bind with the particles on the surface of the teeth, such as food debris, tannins from beverages like tea and coffee, and other external stains, make it an effective ingredient for tooth stain removal [26]. Adding activated charcoal to toothpaste enables its potential as a tooth-whitening agent by removing extrinsic tooth stains. Anecdotal reports and advertisements often claim the superior stain removal capability of activated charcoal over traditional toothpaste, which may contain harsher chemicals. A recent study by Sultan MS, 2024, reported that activated charcoal-based mouthwash had the potential to improve the color of enamel without causing an adverse effect on the surface roughness [27]. Another study by Farghal NS et al., 2024, reported that a combination of charcoal-infused toothbrush and activated-charcoal toothpaste was useful in retaining the gloss of certain restorative materials [28]. However, another study by Balhaddad A. A. et al., 2024 reported that the use of charcoal and whitening toothpastes increased enamel roughness, particularly with long-term use [29].

Among the available over-the-counter and professional teeth-whitening products, hydrogen peroxide and carbamide peroxide are the most commonly used ingredients. Due to their ability to break down tooth stains effectively, they have been integrated into toothpaste formulations for at-home use. Carey CM, 2014, explained that hydrogen peroxide penetrates the enamel and breaks down chromogenic compounds through an oxidation process, which can facilitate teeth whitening [30]. On the other hand, carbamide peroxide is a compound that breaks down into hydrogen peroxide and urea upon contact with moisture in the mouth. The slow release of hydrogen peroxide makes carbamide peroxide an ideal candidate for at-home whitening products, as it can remain active for longer durations compared to hydrogen peroxide alone [31]. In our study, the hydrogen peroxide-based toothpastes Colgate Optic White Expert (group 4) and Colgate Optic White Expert Complete (group 5) showed good whitening effects similar to those from the charcoal-based Herbal Expert Whitening toothpaste (group 1). However, the carbamide peroxide-based toothpastes Close Up White Now Gold (group 2) and Crest 3D White Deluxe (group 3) showed less effectiveness compared to the hydrogen peroxide-based toothpastes. As carbamide peroxide requires a longer time to facilitate a whitening effect, 2 min of brushing might not be enough to obtain the optimal whitening effect [26].

Unlike other toothpastes, Sensodyne True White (group 6) contains sodium tripolyphosphate as an active ingredient for teeth whitening. Sodium tripolyphosphate (STP) chelates metal ions, like calcium and magnesium, and breaks the interaction between stain molecules and the tooth enamel, preventing the attachment of pigments, like tannins from coffee and tea, to the tooth surface [27]. STP also acts as an anti-redeposition agent, meaning that once stains are lifted from the enamel, it prevents them from re-attaching. This helps to remove and prevent surface stains over time and acts gently on the enamel, making it suitable for people with sensitive teeth [28]. Another active ingredient in Sensodyne True White is titanium dioxide. Titanium dioxide is a highly refractive material. When applied to the surface of the teeth, its particles reflect and scatter light. This optical property creates the illusion of whiter, brighter teeth by reflecting white light off their surface. This does not change the intrinsic color of the teeth but enhances their appearance. The incorporation of titanium dioxide with conventional or natural extract-based tooth bleaching gel can enhance the whitening efficiency and enamel surface properties [29,30]. In our study, the combination of STP and titanium oxide in Sensodyne True White demonstrated its potential as a whitening toothpaste. However, its efficacy was less than that of the active charcoal and hydrogen peroxide-based toothpastes.

In the case of Close Up White Now (group 7), blue covarine (CI 74160) is the active ingredient to facilitate teeth whitening. Blue covarine is a pigment that adheres to the surface of teeth. It creates a temporary optical effect by depositing a thin blue film on the enamel, which counteracts yellow tones on the teeth. The blue tint shifts the way light reflects off the teeth, making them appear whiter immediately after brushing. However, it does not remove stains or change the underlying color of the teeth. A randomized clinical trial by Schlafer S. et al., 2021 reported that a single brushing with a blue covarine-containing toothpaste was unable to improve teeth whiteness [31]. In our study, Close Up White after 2 weeks of the staining–brushing cycle showed some whitening effect; however, it was significantly lower than that of the other groups. Other ingredients incorporated into toothpaste, like titanium dioxide (CI 77891), hydrated silica, or other mild abrasives, helps with stain removal over time.

In our study, the group 8 specimens, which were treated with Crest 3D White Pearl Glow, showed the least whitening effect after 2 weeks of the staining–brushing cycle. The pearl powder incorporated into the toothpaste adds mechanical abrasive action to help remove enamel staining. Although it contains hydrogen peroxide in its composition, studies that have tested the whitening efficiency of Crest 3D White Pearl Glow or other pearl powder-based toothpaste are lacking.

The present study shows comparative outcomes of whitening toothpastes with different active ingredients. The toothpastes used in this study are the most commonly used and easily available whitening toothpaste on the market. Despite containing similar active ingredients, their whitening outcomes may due to different concentrations and formulations. However, the clinical outcomes of such products need to be verified by a standard clinical trial. Although these over-the-counter toothpastes showed the potential of teeth whitening, their frequent and long-term should be restricted to avoid possible side effects [32,33]. In-office and at-home professional teeth whitening is recommended for individuals seeking better whitening outcomes [16,34].

## 5. Conclusions

Within the limitations of the in vitro experiment model, the current study demonstrates that the over-the-counter toothpastes used in this study have the potential for teeth whitening. The degrees of teeth whitening of these toothpastes may vary based on their active ingredients and compositions. The activated charcoal and hydrogen peroxide-based toothpastes had the strongest tooth-whitening capability.

## Figures and Tables

**Figure 1 dentistry-13-00045-f001:**
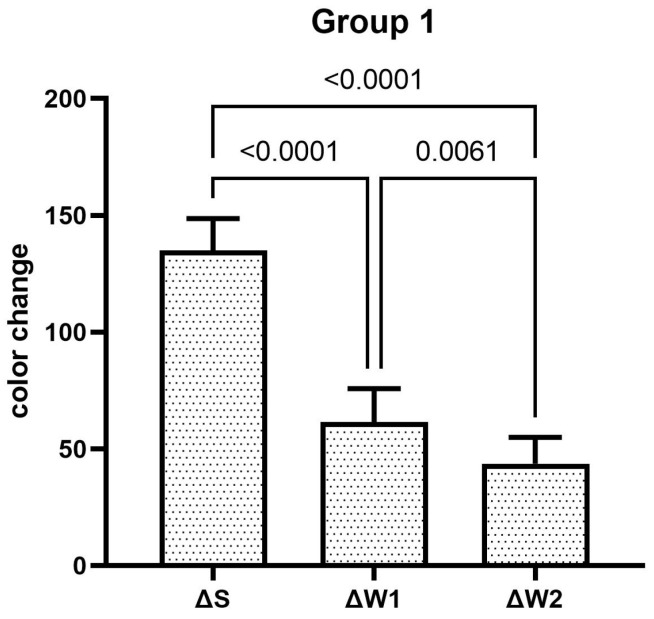
The mean color changes in group 1 after staining, 1 week of the staining–brushing cycle, and 2 weeks of the staining–brushing cycle.

**Figure 2 dentistry-13-00045-f002:**
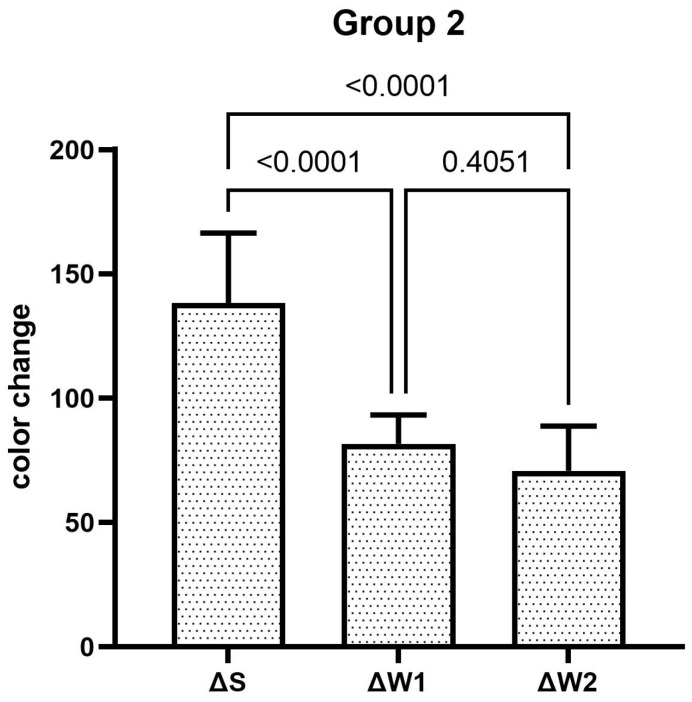
The mean color changes in group 2 after staining, 1 week of the staining–brushing cycle, and 2 weeks of the staining–brushing cycle.

**Figure 3 dentistry-13-00045-f003:**
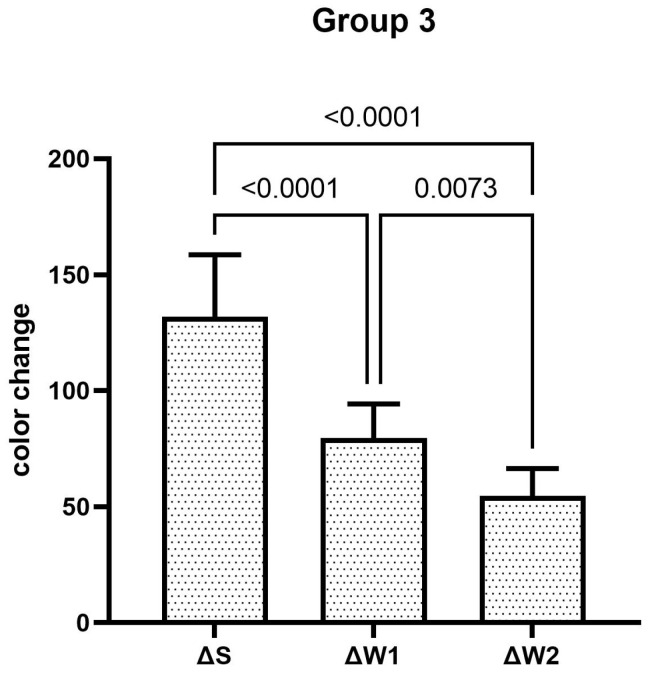
The mean color changes in group 3 after staining, 1 week of the staining–brushing cycle, and 2 weeks of the staining–brushing cycle.

**Figure 4 dentistry-13-00045-f004:**
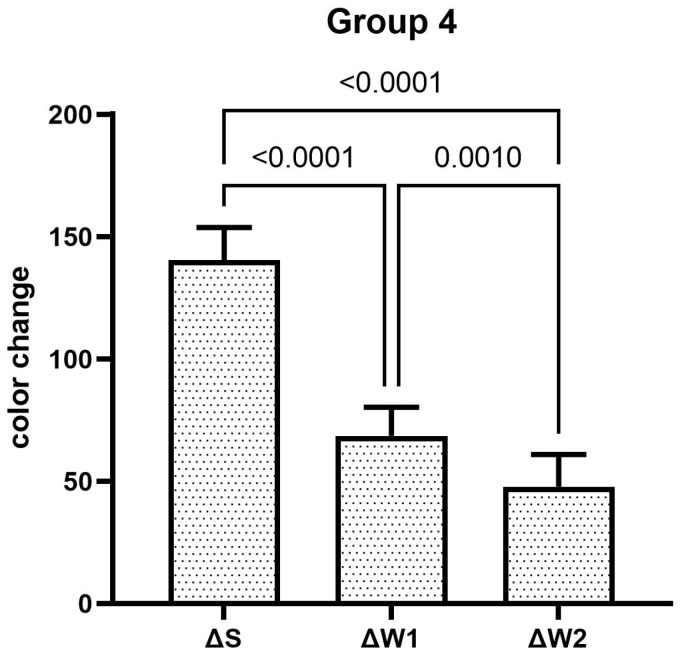
The mean color changes in group 4 after staining, 1 week of the staining–brushing cycle, and 2 weeks of the staining–brushing cycle.

**Figure 5 dentistry-13-00045-f005:**
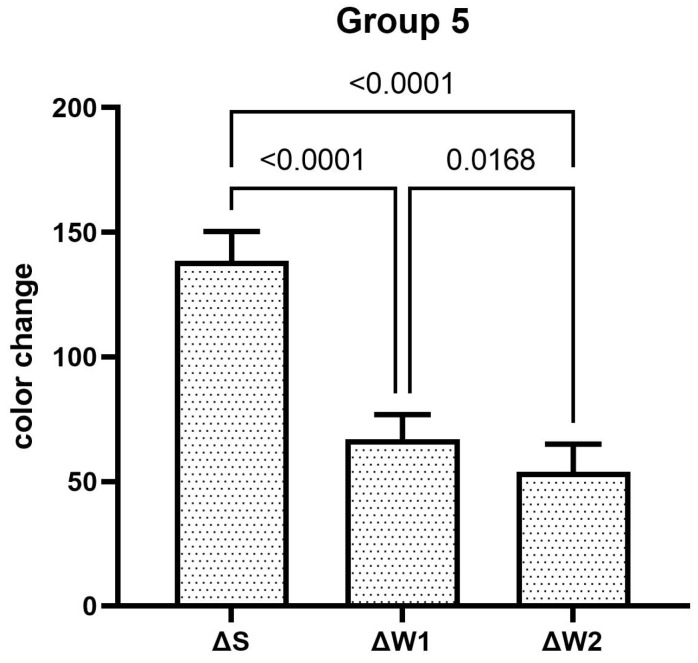
The mean color changes in group 5 after staining, 1 week of the staining–brushing cycle, and 2 weeks of the staining–brushing cycle.

**Figure 6 dentistry-13-00045-f006:**
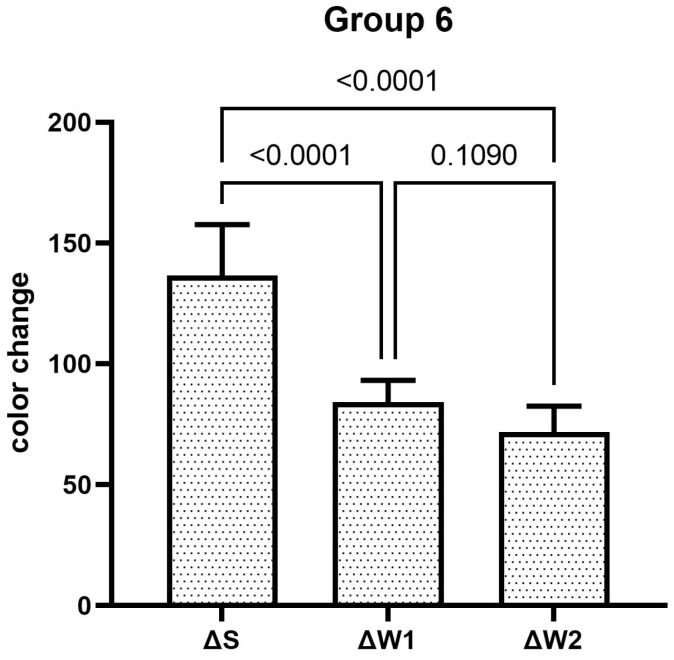
The mean color changes in group 6 after staining, 1 week of the staining–brushing cycle, and 2 weeks of the staining–brushing cycle.

**Figure 7 dentistry-13-00045-f007:**
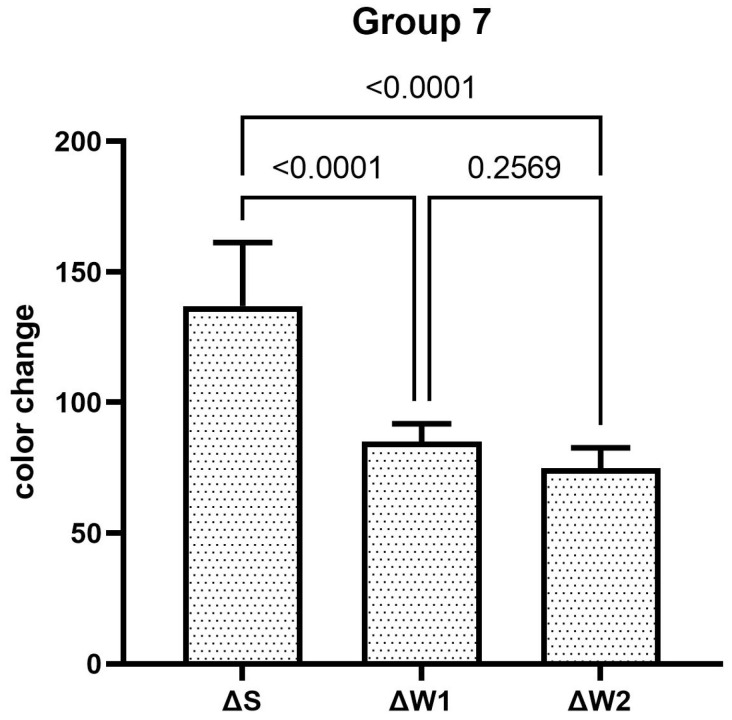
The mean color alteration in group 7 after staining, 1 week of brushing cycle, and 2 weeks of brushing cycle.

**Figure 8 dentistry-13-00045-f008:**
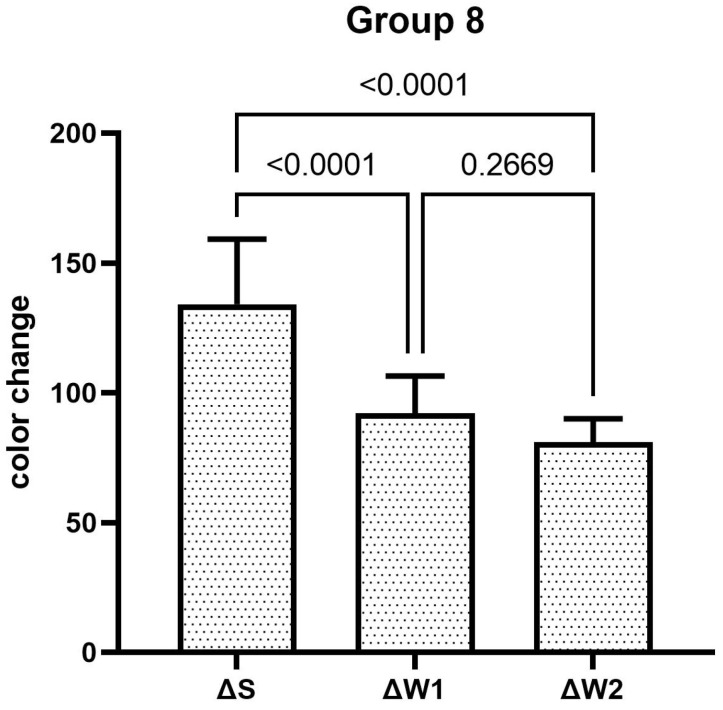
The mean color changes in group 8 after staining, 1 week of the staining–brushing cycle, and 2 weeks of the staining–brushing cycle.

**Figure 9 dentistry-13-00045-f009:**
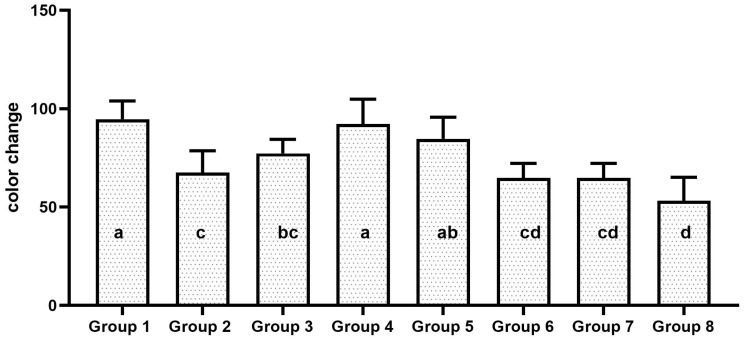
The mean color alteration of all the experimental group after 2 weeks of brushing cycle. Groups identified with same alphabet are statistically insignificant.

**Table 1 dentistry-13-00045-t001:** Group distribution and active ingredients of the toothpastes used in the experiment.

Group No.	Product Name and Manufacturer	Active Ingredients
1	Dabur Herbal Activated Charcoal Toothpaste (Dabur, Dubai, UAE)	Activated charcoal
2	Closeup White Now Gold (Unilever, Egypt)	Hydrogen peroxide and carbamide peroxide
3	Crest 3D White Deluxe (Procter & Gamble, Germany)	Hydrogen peroxide, carbamide peroxide
4	Colgate Optic White Expert (Colgate-Palmolive, Poland)	Hydrogen peroxide
5	Colgate Optic White Expert Complete (Colgate-Palmolive, Poland)	Hydrogen peroxide
6	Sensodyne True White (Gulf Center Cosmetics, UAE)	Sodium tripolyphosphate, titanium dioxide
7	Closeup White Now (Unilever, Egypt)	Blue covarine, titanium dioxide
8	Crest 3D White Pearl Glow (Procter & Gamble, Germany)	Hydrogen peroxide

**Table 2 dentistry-13-00045-t002:** Mean color difference and SD of each experimental group after staining (ΔS), after 1st week of staining–brushing cycle (ΔW1), and after 2nd week of staining–brushing cycle (ΔW2).

	Group 1 (Mean ± SD)	Group 2 (Mean ± SD)	Group 3 (Mean ± SD)	Group 4 (Mean ± SD)	Group 5 (Mean ± SD)	Group 6 (Mean ± SD)	Group 7 (Mean ± SD)	Group 8 (Mean ± SD)
ΔS	134.99 ± 13.6	138.20 ± 28.2	131. 87 ± 26.7	140.49 ± 13.2	138.56 ± 11.8	136.58 ± 21.0	136.91 ± 24.3	134.21 ± 25.0
ΔW1	61.51 ± 14.2	81.56 ± 11.6	79.65 ± 14.6	68.58 ± 11.6	66.95 ± 9.8	84.20 ± 8.9	84.90 ± 6.8	92.25 ± 14.2
ΔW2	43.72 ± 11.2	70.69 ± 18.0	54.64 ± 11.7	47.84 ± 13.1	53.88 ± 11.1	71.82 ± 10.6	74.87 ± 7.8	81.0 ± 8.9

## Data Availability

The raw data supporting the conclusions of this article will be made available by the corresponding author upon request.

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
