# Peer review of "Comparative Analysis of Whitening Outcomes of Over-the-Counter Toothpastes: An In Vitro Study"

_dentistry, 2025, doi:10.3390/dj13020045_

Round 1

Reviewer 1 Report

Comments and Suggestions for Authors

The number of teeth included per group is not indicated (neither in the abstract nor throughout the text).

It seems that there is a lack of precision in taking the colour of the samples. How do you know that you always took the colour at the same place on the tooth?

Although another article is cited to indicate the staining and brushing methodology, it is not fully explained in this one. It should be explained better, as the cited article has different objectives than this one.

Group 8 shows a lower effect than the other groups. However, it shows the same active ingredient in table 1 as groups 4 and 5, which show a similar effect to activated charcoal. How can this be explained?

Finally, I suggest the addition of a control group

  •  

Author Response

​The author appreciates the valuable comments of the reviewer. The comments were really helpful to improve the quality of the manuscript.

Comment: The number of teeth included per group is not indicated (neither in the abstract nor throughout the text).

Reply: The number of specimens in each group (n=4) has been added in the abstract and in the main text.    

Comment: It seems that there is a lack of precision in taking the colour of the samples. How do you know that you always took the colour at the same place on the tooth?

Reply: Thank you for the critical question. Accuracy of the color chromometer used in this study has been explain in our previously published article. It has been included in the discussion. Regarding specimens, as explained in the methodology specific surface areas were flattened and polished with 1000-2000 grid SIC which precise the area of color measurement. Finally, for accuracy, each reading was repeated 3 times.       

Comment: Although another article is cited to indicate the staining and brushing methodology, it is not fully explained in this one. It should be explained better, as the cited article has different objectives than this one.

Reply: Thank you for the comment. The details and rationale of the brushing cycle used in this are explained in the first paragraph of the discussion. The citation referred by the reviewer, is for the composition of artificial saliva used in this study.  

Comment: Group 8 shows a lower effect than the other groups. However, it shows the same active ingredient in table 1 as groups 4 and 5, which show a similar effect to activated charcoal. How can this be explained?

Reply: Thank you for highlighting another critical aspect of the current study. We have added an explanation of this in the discussion of the revised manuscript.

Comment: Finally, I suggest the addition of a control group

Reply: Thank you for another valuable comment. In our study, we compared the whitening effect of each toothpaste by measuring the ΔE. Here stained specimens are considered as baseline/ control, the brushing cycle is the intervention, and color change after the busing cycle compared to the staining condition is the outcome. We intentionally did not include any toothpaste as a control because mechanical abrasion with a toothbrush with or without any toothpaste will show a certain degree of whitening that might mislead the result.  

Reviewer 2 Report

Comments and Suggestions for Authors

While the manuscript addresses a compelling issue, there are several concerns about the study.

Specific comments are noted below:

Title

- The title could be more concise and engaging. It currently lacks clarity and precision in conveying the scope and methods of the study.

- Ensure that the title explicitly highlights the key elements of the study, such as the in vitro nature and comparative evaluation of whitening outcomes.

Abstract

- The objectives should be clearer and more concise. Avoid redundant phrases or overly broad statements.

- The methods section in the abstract lacks detail about the specific materials and processes used.

- The results section needs better organization and clearer explanation of the statistical outcomes, including significance levels and comparisons between groups.

- The conclusions should more directly address the broader implications or potential applications of the findings.

Keywords

- The keywords include a mix of specific technical terms and general descriptors. Consider focusing on terms most relevant to the study for better indexing and discoverability.

- Make sure they correspond to MeSH terms

Introduction

- The introduction has repetitive statements about the societal and aesthetic importance of white teeth. Streamline to avoid redundancy.

- There is an overemphasis on cultural and psychological aspects of teeth whitening, which could detract from the scientific focus of the study.

- Some background information (e.g., details about enamel and dentin) is presented in excessive detail and could be condensed.

- The rationale for the study is mentioned but not strongly justified. Clearly state the gap in knowledge or the specific problem the study aims to address.

- The objectives are scattered and need to be stated more cohesively at the end of the introduction, with a clear link to the methods used.

Materials and Methods

- Ensure that all ethical considerations and approvals, including compliance with ARRIVE guidelines, are explicitly detailed to avoid redundancy and to clarify adherence.

- Provide more precise details on the storage and handling of specimens, including storage conditions before thawing, and describe how the enamel polishing process ensures consistent results.

- Clarify the rationale behind using the specific dental colorimeter and CIE color space parameters. Additionally, ensure the process of averaging multiple measurements is explicitly described.

- The description of the staining process and the formula for color differences needs further elaboration to ensure reproducibility. Cite references appropriately and ensure the method is sufficiently detailed for independent validation.

- Specify the criteria and method used for random allocation of specimens into groups to ensure transparency and eliminate potential bias.

- Clarify the parameters for "constant speed and pressure" during brushing, and ensure the description of the artificial saliva solution is complete, including its preparation method.

- Provide a detailed rationale for the choice of statistical tests, describe the parameters used for significance determination, and justify the use of specific post hoc analyses.

- The placement and explanation of formulas (e.g., CIELab formulas) need improvement for readability and comprehension. Ensure all variables are defined clearly.

- Provide further details on the software and tools used, including specific settings or modifications applied during statistical and graphical analysis.

Results

- The description of the statistical analysis lacks detailed numerical data, such as means, standard deviations, or confidence intervals, which are essential for supporting claims. Include these to strengthen the results.

- Specify the exact comparisons being made between groups. Avoid overly general statements and clearly highlight significant differences and trends.

- The figures referenced (Figure 1-8 and Figure 9) need to be clearly described in the text, including what each figure represents, its significance, and how it supports the findings.

- The results for groups with similar trends (e.g., Group 1, 3, 4, and 5 versus Group 2, 6, 7, and 8) are repeated without adding value. Consolidate findings where appropriate to improve readability.

- Terms like "whitening effect," "color alteration," and "tooth discoloration" are used interchangeably without clear definitions. Ensure consistent and precise terminology throughout.

- Avoid prematurely interpreting results in the Results section. Descriptions should focus solely on presenting findings, leaving interpretation to the Discussion section.

- Ensure that the figures mentioned are logically integrated into the narrative, with a clear explanation of their relevance to the results.

- For results deemed statistically insignificant (e.g., whitening effect after the second week for some groups), provide context or possible explanations in the Discussion section.

- The group distribution in the results is not adequately linked to the information in the materials and methods (e.g., Table 1). Ensure that the groups are clearly identifiable and consistent across sections.

- Statements such as "all 8 toothpastes... had a significant effect" lack specificity about the magnitude of the effects. Provide nuanced descriptions of the findings.

Discussion

- The discussion relies heavily on general claims without sufficient reference to comparable studies or broader scientific context. Expand on how the findings align or deviate from existing literature.

- The discussion does not critically address limitations in the experimental design (e.g., in-vitro conditions vs. real-world application). Consider exploring the implications of these limitations more deeply.

- While the discussion provides mechanisms of action for active ingredients, some are overly simplified or speculative. Clarify the evidence supporting these mechanisms and avoid overgeneralizations.

- The comparison between the efficacy of different toothpaste formulations lacks depth. Discuss why certain ingredients performed better or worse and the potential implications of these findings.

- The discussion lacks sufficient detail about the statistical significance and its practical relevance. Include a clearer interpretation of the statistical results in the context of real-world application.

- The section is lengthy and occasionally repetitive. Streamline the narrative to focus on the key findings and their implications.

- While limitations are mentioned, the study lacks suggestions for future research to address gaps or validate findings under in-vivo conditions.

- Avoid overemphasizing the benefits of specific ingredients or products without adequately discussing potential risks, such as enamel erosion and sensitivity from abrasive materials.

Conclusions 

- The conclusions are broad and do not adequately emphasize the study’s main contributions or practical applications. Provide specific takeaways for both clinical and consumer use.

- Some terms (e.g., "bruising" instead of "brushing") and phrases are inconsistent or imprecise. Review language for accuracy and professionalism.

Comments on the Quality of English Language

Moderate editing

Author Response

While the manuscript addresses a compelling issue, there are several concerns about the study.

Specific comments are noted below:

Comment:  The title could be more concise and engaging. It currently lacks clarity and precision in conveying the scope and methods of the study.

- Ensure that the title explicitly highlights the key elements of the study, such as the in vitro nature and comparative evaluation of whitening outcomes.

Reply: Thank you for the comment. The title has been revised as per the suggestion

Abstract

Comment:  The objectives should be clearer and more concise. Avoid redundant phrases or overly broad statements.

Reply: Thank you for the comment. The objective has been revised as per the comment.

Comment:  The methods section in the abstract lacks detail about the specific materials and processes used.

Reply: Thank you for the comment. The methodology has been revised as per the comment

Comment: - The results section needs better organization and clearer explanation of the statistical outcomes, including significance levels and comparisons between groups.

Reply: Thank you for the comment. The result has been revised as per the suggestion

Comment: The conclusions should more directly address the broader implications or potential applications of the findings.

Reply: Thank you for the comment. The conclusion has been revised

Comment: The keywords include a mix of specific technical terms and general descriptors. Consider focusing on terms most relevant to the study for better indexing and discoverability. Make sure they correspond to MeSH terms

Reply: Thank you for the suggestion. The currently used keywords include the active ingredients, methods and outcome. The authors are of the opinion that it is necessary to keep the same.    

Comment: The introduction has repetitive statements about the societal and aesthetic importance of white teeth. Streamline to avoid redundancy. There is an overemphasis on cultural and psychological aspects of teeth whitening, which could detract from the scientific focus of the study.

Reply: Thank you for the comment. The statements have been revised

Comment: Some background information (e.g., details about enamel and dentin) is presented in excessive detail and could be condensed.

Reply: Thank you for the comment. This section has been revised

Comment: The rationale for the study is mentioned but not strongly justified. Clearly state the gap in knowledge or the specific problem the study aims to address.

Reply: Thank you for the comment. This section has been revised

Comment: The objectives are scattered and need to be stated more cohesively at the end of the introduction, with a clear link to the methods used.

Reply: Thank you for the comment. This section has been revised

Comment:  Ensure that all ethical considerations and approvals, including compliance with ARRIVE guidelines, are explicitly detailed to avoid redundancy and to clarify adherence.

Reply: Thank you for the comment. The statement regarding ARRIVE guideline has been removed as no live animal was used in this study.

Comment: Provide more precise details on the storage and handling of specimens, including storage conditions before thawing, and describe how the enamel polishing process ensures consistent results.

Reply: Thank you for the comment. Suggested statement has been added

 Comment: Clarify the rationale behind using the specific dental colorimeter and CIE color space parameters. Additionally, ensure the process of averaging multiple measurements is explicitly described.

Reply: Thank you for the comment. This section has been revised.

Comment: The description of the staining process and the formula for color differences needs further elaboration to ensure reproducibility. Cite references appropriately and ensure the method is sufficiently detailed for independent validation.

Reply: Thank you for the comment. This section has been revised.

Comment: Specify the criteria and method used for random allocation of specimens into groups to ensure transparency and eliminate potential bias.

Reply: Thank you for the comment. The specimens were allocated to different groups using a manual randomization method.

Comment: Clarify the parameters for "constant speed and pressure" during brushing, and ensure the description of the artificial saliva solution is complete, including its preparation method.

Reply: Thank you for the comment. The section has been revised as per the suggestion

Comment: Provide a detailed rationale for the choice of statistical tests, describe the parameters used for significance determination, and justify the use of specific post hoc analyses.

Reply: Thank you for the comment. The section has been revised as per the suggestion

Comment: The placement and explanation of formulas (e.g., CIELab formulas) need improvement for readability and comprehension. Ensure all variables are defined clearly.

Reply: Thank you for the comment. The section has been revised as per the suggestion

Comment: Provide further details on the software and tools used, including specific settings or modifications applied during statistical and graphical analysis.

Reply: The software details have been provided in the version 1. The author requests the reviewer to be specific on what kind of detail is expected to add.

 Comment: The description of the statistical analysis lacks detailed numerical data, such as means, standard deviations, or confidence intervals, which are essential for supporting claims. Include these to strengthen the results.

Reply: Thank you for the comment. The numerical data has been added in the result section.

Comment: Specify the exact comparisons being made between groups. Avoid overly general statements and clearly highlight significant differences and trends.

Reply: Thank you for the comment. The section has been revised as per the suggestion

Comment: The figures referenced (Figure 1-8 and Figure 9) need to be clearly described in the text, including what each figure represents, its significance, and how it supports the findings.

Reply: The legend of each figure has been provided in version 1. The author requests the reviewer to be specific about what is missing.

Comment: The results for groups with similar trends (e.g., Group 1, 3, 4, and 5 versus Group 2, 6, 7, and 8) are repeated without adding value. Consolidate findings where appropriate to improve readability.

Reply: Thank you for the comment. A numeric value has been added

Comment: Terms like "whitening effect," "color alteration," and "tooth discoloration" are used interchangeably without clear definitions. Ensure consistent and precise terminology throughout.

Reply: Thank you for the comment. The section has been revised as per the suggestion

Comment: Avoid prematurely interpreting results in the Results section. Descriptions should focus solely on presenting findings, leaving interpretation to the Discussion section.

Reply: Thank you for the comment. The section has been revised as per the suggestion

Comment: Ensure that the figures mentioned are logically integrated into the narrative, with a clear explanation of their relevance to the results.

Reply: Thank you for the comment. The section has been revised as per the suggestion

Comment: For results deemed statistically insignificant (e.g., whitening effect after the second week for some groups), provide context or possible explanations in the Discussion section.

Reply: Thank you for the comment. The section has been revised as per the suggestion

Comment: The group distribution in the results is not adequately linked to the information in the materials and methods (e.g., Table 1). Ensure that the groups are clearly identifiable and consistent across sections.

Reply: Thank you for the comment. The section has been revised as per the suggestion

Comment: Statements such as "all 8 toothpastes... had a significant effect" lack specificity about the magnitude of the effects. Provide nuanced descriptions of the findings.

Reply: Thank you for the comment. The section has been revised as per the suggestion

Discussion

Comment: The discussion relies heavily on general claims without sufficient reference to comparable studies or broader scientific context. Expand on how the findings align or deviate from existing literature.

Reply: Thank you for the comment. The section has been revised as per the suggestion

Comment: The discussion does not critically address limitations in the experimental design (e.g., in-vitro conditions vs. real-world application). Consider exploring the implications of these limitations more deeply.

Reply: Thank you for the comment. The section has been revised as per the suggestion

Comment: While the discussion provides mechanisms of action for active ingredients, some are overly simplified or speculative. Clarify the evidence supporting these mechanisms and avoid overgeneralizations.

Reply: Thank you for the comment. The section has been revised as per the suggestion

Comment: The comparison between the efficacy of different toothpaste formulations lacks depth. Discuss why certain ingredients performed better or worse and the potential implications of these findings.

Reply: Thank you for the comment. The section has been revised as per the suggestion

Comment: The discussion lacks sufficient detail about the statistical significance and its practical relevance. Include a clearer interpretation of the statistical results in the context of real-world application.

Reply: Thank you for the comment. The section has been revised as per the suggestion

Comment: The section is lengthy and occasionally repetitive. Streamline the narrative to focus on the key findings and their implications.

Reply: Thank you for the comment. The section has been revised as per the suggestion

Comment: While limitations are mentioned, the study lacks suggestions for future research to address gaps or validate findings under in-vivo conditions.

Reply: Thank you for the comment. The section has been revised as per the suggestion

Comment: Avoid overemphasizing the benefits of specific ingredients or products without adequately discussing potential risks, such as enamel erosion and sensitivity from abrasive materials.

Reply: Thank you for the comment. The section has been revised as per the suggestion

Conclusions

Comment: The conclusions are broad and do not adequately emphasize the study’s main contributions or practical applications. Provide specific takeaways for both clinical and consumer use.

Reply: Thank you for the comment. The section has been revised as per the suggestion

Comment: Some terms (e.g., "bruising" instead of "brushing") and phrases are inconsistent or imprecise. Review language for accuracy and professionalism.

Reply: Thank you for the comment. The section has been revised as per the suggestion

Reviewer 3 Report

Comments and Suggestions for Authors

The authors evaluated the whitening efficacy of eight over-the-counter toothpastes using an in vitro stain-brushing model. Results indicated significant variability, with activated charcoal and hydrogen peroxide-based products performing best. The authors concluded that these were effective for extrinsic stains, long-term safety and enamel preservation remain concerns. The reviewer has a few comments.

Overall Comments

The English language is clear and professional, but there are occasional grammatical errors.

- Instances of "bruising" instead of "brushing".

- Be consistent with "brushing cycle" vs "tooth-brushing cycle".

Introduction

The introduction provides a sufficient overview of the significance of tooth whitening and the role of over-the-counter whitening toothpastes. However, some relevant areas, such as more comprehensive prior research on activated charcoal and hydrogen peroxide in toothpastes, are lightly addressed. 

- Reduce redundancy in discussing the social perceptions of tooth whiteness.

- Provide a more detailed review of prior studies on specific active ingredients and their whitening mechanisms.

- Clarify the novelty of the study compared to existing research.

Research Design

The research design, which simulates brushing cycles and staining with tea-based solutions, is appropriate for evaluating whitening efficacy. However, there is limited explanation of why specific groups of toothpastes were chosen.

- Justify the selection of the toothpaste brands and ingredients tested in the study.

- Provide a rationale for the two-week brushing cycle duration. Consider whether longer cycles might yield more realistic insights.

Methods

The methods are described in adequate detail, but some steps lack precision.

- Include details on the consistency of pressure and speed during brushing cycles.

- Provide the specific composition of artificial saliva and cite its relevance for mimicking oral conditions.

- Elaborate on the statistical methods used.

Results

The results are presented with sufficient data and clear statistical significance, but some figures and descriptions are repetitive.

Discussion

The discussion contextualises the results well and compares them with previous studies. However, it occasionally repeats findings rather than critically analysing them.

- Highlight the limitations of in vitro studies and suggest future directions, such as clinical trials.

- Provide a deeper analysis of why carbamide peroxide performed poorly compared to hydrogen peroxide.

Conclusion

The conclusions are supported by the results but could be more succinct. The potential limitations of the whitening toothpastes are not emphasised enough.

- Address limitations and highlight the need for further studies.

Author Response

The authors evaluated the whitening efficacy of eight over-the-counter toothpastes using an in vitro stain-brushing model. Results indicated significant variability, with activated charcoal and hydrogen peroxide-based products performing best. The authors concluded that these were effective for extrinsic stains, long-term safety and enamel preservation remain concerns. The reviewer has a few comments.

Reply: Thank you for the in-depth review of the article. The comments were useful in improving the quality of the manuscript

Overall Comments

Comment: The English language is clear and professional, but there are occasional grammatical errors.

Reply: Thank you for the comment. The grammatical error has been revised carefully.

Comment: Instances of "bruising" instead of "brushing".

Reply: Thank you for the comment. The typo has been revised carefully

Comment: Be consistent with "brushing cycle" vs "tooth-brushing cycle".

Reply: Thank you for the comment. The typo has been revised carefully

Introduction

Comment: The introduction provides a sufficient overview of the significance of tooth whitening and the role of over-the-counter whitening toothpastes. However, some relevant areas, such as more comprehensive prior research on activated charcoal and hydrogen peroxide in toothpastes, are lightly addressed.

Reply: Thank you for the comment. The section has been revised as per the suggestion

Comment: Reduce redundancy in discussing the social perceptions of tooth whiteness.

Reply: Thank you for the comment. The section has been revised as per the suggestion

Comment: Provide a more detailed review of prior studies on specific active ingredients and their whitening mechanisms.

Reply: Thank you for the comment. The section has been revised as per the suggestion

Comment: Clarify the novelty of the study compared to existing research.

Reply: Thank you for the comment. The section has been revised as per the suggestion

Research Design

Comment: The research design, which simulates brushing cycles and staining with tea-based solutions, is appropriate for evaluating whitening efficacy. However, there is limited explanation of why specific groups of toothpastes were chosen.

Reply: Thank you for the nice comment. This study was conducted in UAE and the toothpaste chosen in this study was based on their availability in most of the supermarkets. They are frequently advertised in the media as well. We have added the same in the manuscript.

Comment: Justify the selection of the toothpaste brands and ingredients tested in the study.

Reply: Thank you for the comment. The section has been revised as per the suggestion

Comment: Provide a rationale for the two-week brushing cycle duration. Consider whether longer cycles might yield more realistic insights.

Reply: Thank you for the comment. The section has been revised as per the suggestion

Methods

Comment: The methods are described in adequate detail, but some steps lack precision.

Reply: Thank you for the comment. The section has been revised as per the suggestion

Comment: Include details on the consistency of pressure and speed during brushing cycles.

Reply: Thank you for the comment. The section has been revised as per the suggestion

Comment:  Provide the specific composition of artificial saliva and cite its relevance for mimicking oral conditions.

Reply: Thank you for the comment. The section has been revised as per the suggestion

Comment: - Elaborate on the statistical methods used.

Reply: Thank you for the comment. The section has been revised as per the suggestion

Results

Comment: The results are presented with sufficient data and clear statistical significance, but some figures and descriptions are repetitive.

Reply: Thank you for the comment. We have a limit expressing the result by bar graph with statistical significance. However, reviewer 2 has asked to add descriptive and numeric data for all the bar graphs. To comply with both the reviewers we have included a table expressing the numeric value of the data and refrain from adding descriptive text of the same.

Discussion

Comment: The discussion contextualises the results well and compares them with previous studies. However, it occasionally repeats findings rather than critically analysing them.

Reply: Thank you for the comment. The section has been revised as per the suggestion

Comment: Highlight the limitations of in vitro studies and suggest future directions, such as clinical trials.

Reply: Thank you for the comment. The limitations of in vitro studies and suggested future direction have been added

Comment: Provide a deeper analysis of why carbamide peroxide performed poorly compared to hydrogen peroxide.

Reply: Thank you for the comment. The section has been revised as per the suggestion

Conclusion

Comment: The conclusions are supported by the results but could be more succinct. The potential limitations of the whitening toothpastes are not emphasised enough.

Reply: Thank you for the comment. The conclusion has been revised as per the suggestion

Comment: Address limitations and highlight the need for further studies.

Reply: Thank you for the comment. The limitation of the study has been added